# Matching between Donors and Ulcerative Colitis Patients Is Important for Long-Term Maintenance after Fecal Microbiota Transplantation

**DOI:** 10.3390/jcm9061650

**Published:** 2020-05-31

**Authors:** Koki Okahara, Dai Ishikawa, Kei Nomura, Shoko Ito, Keiichi Haga, Masahito Takahashi, Tomoyoshi Shibuya, Taro Osada, Akihito Nagahara

**Affiliations:** Department of Gastroenterology, Juntendo University School of Medicine, 2-1-1 Hongo, Bunkyo-ku, Tokyo 113-8421, Japan; k-okahara@juntendo.ac.jp (K.O.); ke-nomura@juntendo.ac.jp (K.N.); soyamada@juntendo.ac.jp (S.I.); khaga@juntendo.ac.jp (K.H.); matakaha@juntendo.ac.jp (M.T.); tomoyosi@juntendo.ac.jp (T.S.); otaro@juntendo.ac.jp (T.O.); nagahara@juntendo.ac.jp (A.N.)

**Keywords:** antibiotic-fecal microbiota transplantation, ulcerative colitis, donor-patient matching

## Abstract

We previously demonstrated that fresh fecal microbiota transplantation (FMT) following triple antibiotic therapy (amoxicillin, fosfomycin, metronidazole (AFM); A-FMT) resulted in effective colonization of Bacteroidetes species, leading to short-term clinical response in ulcerative colitis (UC). Its long-term efficacy and criteria for donor selection are unknown. Here, we analyzed the long-term efficacy of A-FMT compared to AFM monotherapy (mono-AFM). AFM was administered to patients with mild to severe UC for 2 weeks until 2 days before fresh FMT. Clinical response and efficacy maintenance were defined by the decrease and no exacerbation in clinical activity index. The population for intention-to-treat analysis comprised 92 patients (A-FMT, *n* = 55; mono-AFM, *n* = 37). Clinical response was observed at 4 weeks post-treatment (A-FMT, 56.3%; mono-AFM, 48.6%). Maintenance rate of responders at 24 months post-treatment was significantly higher with A-FMT than mono-AFM (*p* = 0.034). Significant differences in maintenance rate according to the age difference between donors and patients were observed. Additionally, sibling FMT had a significantly higher maintenance rate than parent–child FMT. Microbial analysis of patients who achieved long-term maintenance showed that some exhibited similarity to their donors, particularly Bacteroidetes species. Thus, A-FMT exhibited long-term efficacy. Therefore, matching between donors and UC patients may be helpful in effectively planning the FMT regimen.

## 1. Introduction

Recent studies have suggested that microbiota is associated with not only digestive diseases but also several other diseases (e.g., liver disease, cardiovascular disease, obesity, allergies, diabetes, mental disorders) [1,2,3,4,5]. Fecal microbiota transplantation (FMT) has been proposed as a microbial therapy for a wide range of other dysbiosis-related diseases such as metabolic syndrome [6], irritable bowel syndrome [2], autism spectrum disorder [7], and inflammatory bowel disease (IBD), considering that the usefulness of FMT for *Clostridioides difficile* infection (CDI) was reported by Van Nood [8,9]. Ulcerative colitis (UC) is considered to be intricately attributable to numerous factors, including genetic [10], immunological, and environmental factors [11], and microbiota is one of these [12,13,14]. Some randomized controlled trials targeting UC have been reported from 2015 to 2019 [15,16,17,18]. Three reports have described the efficacy of FMT, administered multiple times, in UC patients [15,16,17]. Although the efficacy of FMT as a short-term treatment for UC has been recognized, the method employed is too cumbersome to be put into practical use. To our knowledge, few studies have examined the therapeutic efficacy of FMT as a long-term treatment for UC [19]. Additionally, several studies have employed volunteer donors, and there is no consensus on the selection of appropriate donors [20]. Increasing the diversity of microbiota is important for inducing remission [21] and matching between donors and recipients may be important when considering stabilized microbiota for prevention of recurrence [22]. Therefore, long-term evaluation of FMT is indispensable and may lead to the formulation of criteria for the selection of appropriate donors. In the present study, our method called A-FMT (FMT following triple antibiotic therapy (amoxicillin, fosfomycin, and metronidazole; AFM)) is simple to use in clinical practice because it entails FMT via colonoscopy once after antibiotic administration for 2 weeks. We had previously reported that a single session of A-FMT efficiently changed the gut microbiota and the change in Bacteroidetes species was associated with the short-term efficacy of the treatment [23,24]. It was proven that patients who showed a therapeutic effect had their gut microbiota successfully transplanted by the donors. In the present study, we analyzed the long-term efficacy of A-FMT compared to AFM monotherapy (mono-AFM) and explored the criteria for a beneficial donor for long-term maintenance (24 months) after A-FMT.

## 2. Materials and Methods

### 2.1. Ethical Considerations

The study protocol was reviewed and approved by the ethics committee of the Juntendo Institutional Review Board at the Juntendo University School of Medicine and the clinical study committee of Juntendo University Hospital (approval numbers 14-017, 15-059, and 16-201). Furthermore, this trial has been registered at http://www.umin.ac.jp/ctr/index-j.htm:UMIN000014152, http://www.umin.ac.jp/ctr/index-j.htm:UMIN000018642, and http://www.umin.ac.jp/ctr/index-j.htm:UMIN000025846. All participants provided written informed consent. All studies were performed at Juntendo University Hospital (Tokyo, Japan).

### 2.2. Patient and Public Involvement

This research was conducted without patient involvement. Patients were not invited to comment on the study design or to contribute to the writing or editing of the article for readability or accuracy and were not consulted to develop relevant patient outcomes or interpret the results.

### 2.3. Patients

A non-randomized controlled study was conducted between July 2014 and March 2017 at Juntendo University Hospital (Tokyo, Japan). The diagnosis of UC was based on standard clinical, endoscopic, and histological findings [25]. The inclusion criteria for this study were as follows: age ≥20 years, diagnosis of active UC with a Lichtiger’s clinical activity index (CAI) score ≥5, or Mayo endoscopic score ≥1. This study included patients with UC who did not achieve clinical remission because of refraction to adequate treatment, including 5-aminosalicylates (5-ASA), immunomodulators, corticosteroids, and/or biologics. In addition, patients with allergies and/or side effects to 5-ASA and biologics were also included, even if their UC was mild. A total of 92 patients were enrolled in this study. Of these patients, 37 were treated with mono-AFM and included in the control group. Meanwhile, 55 patients were treated with A-FMT. The patients chose the mono-AFM or A-FMT regimen by themselves. The patients who could not find an appropriate donor candidate chose the mono-AFM regimen. Additionally, some patients chose mono-AFM owing to its convenience and its proven efficacy.

The exclusion criteria were the presence of cytomegalovirus colitis, current serious disease, pregnancy, participation in other clinical studies, or antibiotic use within 3 months prior to participation in this study.

### 2.4. Donors

Spouses and relatives were selected as donors by the patients. Donors needed to be ≥20 years old and in good health. Donor candidates provided information regarding their medical history, current diseases, travel history, sexual behavior, recent gastrointestinal tract infections, and defecation habits.

Donor candidates who had used antibiotics within 3 months before participating in the study were not included. The stool and blood samples of the donors were collected at the time of their visit to the hospital. Blood and feces samples of donors were checked using the Amsterdam Protocol [26,27,28]. The donors with the serologic presence of hepatitis B and C viruses, human immunodeficiency virus, cytomegalovirus, and syphilis were excluded from the study. The feces samples were tested for the presence of *Clostridioides difficile*, enterohemorrhagic *Escherichia coli*, *Salmonella*, *Shigella*, *Yersinia*, *Campylobacter*, parasites, and helminths. If this screening test was positive, the donor candidate was excluded from the study.

### 2.5. Protocol for AFM Monotherapy

Patients received a combination antibiotic regimen consisting of oral amoxicillin (1500 mg/day), fosfomycin (3000 mg/day), and metronidazole (750 mg/day) for 2 weeks. After AFM monotherapy, the patients were permitted to continue the ongoing treatment and were regulated in terms of probiotic intake during the study period. Change in drug dosage or the initiation of new treatment was not allowed.

### 2.6. Protocol for A-FMT

We have already reported the method of A-FMT therapy [23,24]. Briefly, the patients received AFM therapy for 2 weeks until 2 days before undergoing fresh FMT. We placed 300–500 mL of the extracted suspension containing the donor’s microbiota in 20 mL syringes. After bowel lavage using a standard polyethylene glycol solution (Moviprep; EA Pharma, Tokyo, Japan), the patients underwent total colonoscopy. Then, 350–500 mL of the diluted and filtered bacterial suspensions were transferred into the patient’s colon, preferably within 6 h after collecting stool samples. About two-thirds of the bacterial suspension was injected into the cecum and ascending colon, and the remaining portion was injected into the transverse colon. After the procedure, scopolamine was administered to patients to slow the intestinal transit and allow sufficient time for the colonization of donor bacteria; following which, the patients had the same restrictions as those in the mono-AFM group.

### 2.7. Clinical Evaluation

For short-term estimation, the clinical features of UC were evaluated using the CAI score before treatment, after antibiotic combination therapy, and at 4 weeks after treatment. After 4 weeks of treatment, clinical response was defined as a CAI score of ≤10 points and a decrease of ≥3 points. Meanwhile, remission was defined as a CAI score of ≤3 points. Disease activity was evaluated at the time of colonoscopy. In the A-FMT group, colonoscopy was conducted at the time of FMT following the AFM administration. The clinical features were observed at 4 weeks after the FMT. In the mono-AFM group, the colonoscopy was performed after AFM administration and clinical features were observed at 4 weeks after the AFM. The endoscopic assessment was performed using the Mayo endoscopic score, particularly the sum of Mayo endoscopic scores [29], which included seven segments of the large intestine (appendicular region, cecum, ascending colon, transverse colon, descending colon, sigmoid colon, and rectum), to determine the overall disease status of patients with UC. For the long-term evaluation of short-term responders, clinical score evaluation was performed at the time of outpatient visit every 1–2 months. Patients who could not visit our hospital as outpatients were regularly interviewed via telephone and mail every 3–6 months. Subsequently, it was confirmed whether their symptoms were exacerbated (estimated by CAI) and/or whether a new treatment was initiated. Patients exhibiting exacerbation of symptoms and/or intensification of treatment, which included an increase in dosage or switching to a new treatment, were defined as being in relapse.

### 2.8. Microbial Analysis

In this study, we used universal *hsp60* primers without adapter sequences. We have already reported the method for microbial analysis involving DNA extraction, amplification by a polymerase chain reaction, preparation of DNA libraries for next-generation sequencing, quality filtering of sequencing reads, and taxonomic analysis based on Bacteroidetes HSP60 sequences [24]. Fecal samples for microbial analysis were collected from 5 out of 10 patients who remained to demonstrate no recurrence for 24 months after undergoing A-FMT and from individual FMT donors. The feces from individual patients and donors were diluted 10-fold in TE buffer (10 mM Tris, 1 mM EDTA (pH 8.0]) and were frozen at −80 °C until use. We observed the relationship of gut microbiota between the patients who had treatment effects and their donors using HSP60 sequences. We analyzed bacterial species with ≥0.1% presence in the donors and post-treatment patients and excluded the unclassified bacterial species.

### 2.9. Statistical Analysis

The results of the clinical efficacy were evaluated by intention-to-treat (ITT) analysis. Differences in the event-free survival rate between the mono-AFM and A-FMT groups were estimated using the Gehan–Breslow–Wilcoxon test and log-rank test. Analysis of factors between donors and patients in A-FMT was conducted using the log-rank test. Differences with *P* values ≤0.05 were considered statistically significant. Chi-square test was used to compare clinical response and clinical remission between the groups. Similarities between Bacteroidetes species in stool samples were calculated using Morisita’s Cλ and Kimoto’s Cπ values, and a dendrogram was constructed from Horn’s R0 values [30,31,32].

### 2.10. Data Availability Statement

All data generated or analyzed during this study are included in this published article and its Appendix A.

## 3. Results

### 3.1. Adverse Events

Among 92 patients treated with AFM, 12 patients (13.0%) showed the exacerbation of diarrhea, rash, and nausea due to antibiotics. However, after the end of the AFM regimen or withdrawal from oral administration (drop-out from the study), all symptoms improved, and no serious drug-related toxicities were observed. Adverse events due to FMT included exacerbation of nausea and prokinetic bowel peristalsis, which were observed in 20 patients (42.6%). Nevertheless, both were transient, and no serious infection or diarrhea was observed.

### 3.2. Short-Term Clinical Evaluation

A total of 92 patients were enrolled in this study between July 2014 and March 2017. Of these patients, 37 were assigned to the mono-AFM group, and 55 were allocated to the A-FMT group. Among 37 patients in the mono-AFM group, five withdrew without waiting for treatment evaluation after 4 weeks because of the adverse effects of antibiotics, exacerbation of other diseases, improvement in symptoms before AFM administration, and no hospital visit. Among 55 patients in the A-FMT group, eight withdrew because of the adverse effects of antibiotics, exacerbation of UC, physician’s decision, and no hospital visit. The clinical characteristics of these 92 patients are summarized in Table 1. The ratio of males to females was significantly higher in the A-FMT group than in the mono-AFM group.

The clinical efficacy of treatment regimens for UC was evaluated using ITT analysis and estimated using the CAI score after 4 weeks of treatment. In the A-FMT group, 31 patients (56.3%) showed a clinical response and 19 patients (34.5%) achieved clinical remission. These rates were higher than the rates observed in the mono-AFM group (clinical response/remission: *n* = 18/6, 48.6%/16.2%) (Figure 1).

The sum of Mayo endoscopic scores of responders in the A-FMT group was significantly lower than that of non-responders (responders: *n* = 31, 5.0 ± 3.6; non-responders: *n* = 16, 7.6 ± 3.1; *p* = 0.02). In contrast, this tendency was not observed in the mono-AFM group (responders: *n* = 18, 6.8 ± 4.1; non-responders: *n* = 14, 5.7 ± 4.5; *p* = 0.47) (Appendix A). Comparison of disease distribution revealed that the response and remission rates in the A-FMT group were significantly higher among patients with proctitis than among patients with left-sided and extensive colitis (left-sided and extensive colitis, *n* = 46; proctitis, *n* = 9; clinical response, *p* = 0.06; clinical remission, *p* = 0.005). No significant differences in the extent of disease were observed in the mono-AFM group (left-sided and extensive colitis, *n* = 33; proctitis, *n* = 4; clinical response, *p* = 1; clinical remission, *p* = 0.52) (Figure 2).

Comparing the efficacy of A-FMT in patients according to ongoing and past medication use, we found that the decrease in the CAI score was significantly lower in users of anti-tumor necrosis factor-α and corticosteroids than in non-users (*p* = 0.03, *p* = 0.009, respectively) (Appendix A). No significant differences in other treatments were observed for the decrease in CAI score. Additionally, no significant difference was observed in the correlation between donor matching and efficacy.

### 3.3. Long-Term Clinical Evaluation

In the A-FMT group, 13 patients (23.6%; ITT analysis) did not experience relapse within 12 months, and 10 (18.2%) patients did not experience relapse within 24 months after undergoing A-FMT. Meanwhile, in the mono-AFM group, six patients (16.2%) did not experience relapse within 12 months, and four patients (10.8%) did not experience relapse within 24 months after AFM administration. Long-term evaluation between two groups was conducted on patients in whom a therapeutic effect was observed at 4 weeks after treatment. One patient in the A-FMT group was excluded because of dysplasia detected by biopsy at the time of FMT; wherein, his clinical symptoms had improved. Therefore, the long-term course was examined in 30 patients belonging to the A-FMT group (Table 2) and 18 patients belonging to the mono-AFM group (Figure 3).

Patients exhibiting exacerbation of symptoms, which led to an increase in the CAI score by ≥1 point, and/or intensification of treatment were defined as being in relapse. The cumulative non-relapse rate after 24 months was significantly higher in the A-FMT group than in the mono-AFM group (*p* = 0.034, Wilcoxon test; Figure 4).

Further, we examined the relationship between factors related to therapeutic effects and donors in the A-FMT group. We divided the patients into two groups according to the age difference between the donor and the patient—namely, the 0–10-year difference group (*n* = 16) and the ≥11-year difference group (*n* = 14)—and subsequently conducted a long-term evaluation. The cumulative non-relapse rate was significantly lower in the ≥11-year difference group than in the 0–10-year difference group (*p* = 0.003, log-rank test; Figure 5a). Until now, the elderly donors were considered to be unsuitable for FMT. However, the FMT employing elderly donors proved to be effective in our study (Table 2, cases #4, #25, and #26). We also divided the patients into the following three groups based on the relationship with the donor: sibling FMT (*n* = 7), parent-child or child-parent FMT (*n* = 13), and other FMT (*n* = 10, including spouse and cousin). We observed that sibling FMT had a significantly higher cumulative non-relapse rate than parent-child FMT (*p* = 0.007, log-rank test; Figure 5b).

### 3.4. Microbial Analysis for Long-Term Study

Bacteroidetes species that existed in donors continued to exist in five patients who remained to demonstrate no recurrence for 24 months (Table 3).

*Bacteroides uniformis* and *Parabacteroides distasonis* were found in all five cases, and Bacteroides dorei was found in 4 out of 5 cases. We showed the change in the composition of Bacteroidetes species between the Bacteroidetes species in 10 samples (donor, patient before A-FMT, at 2 weeks and 1, 2, 3, 5, 6, 12, and 24 months after A-FMT) for case #11 (Figure 6a). Ten Bacteroidetes species existed for 24 months (Table 3) and were more similar to the donor than to the recipient’s bacterial composition before the therapy. The patient was in remission (both clinical and endoscopic) for 24 months after receiving a single A-FMT regimen (Figure 6b). These results suggest that there was an improvement in microbiota colonized from a healthy donor, along with clinical improvement after undergoing A-FMT. However, such a clear tendency was not confirmed in other cases (Appendix A).

(1)Baseline endoscopic appearance of the descending colon with active colitis; clinical activity index (CAI) score = 14, endoscopic subscore = 3.(2)Endoscopic appearance at the end of 1 month after A-FMT; CAI score = 4, endoscopic subscore = 1.(3)Endoscopic appearance at the end of 6 months after A-FMT; CAI score = 1, endoscopic subscore = 1.(4)Endoscopic appearance at the end of 24 months after A-FMT; CAI score = 1, endoscopic subscore = 1.

The patient remained in clinical remission at the final follow-up of the study (i.e., 24 months after A-FMT).

## 4. Discussion

During the short-term evaluation, the response and remission rates in the present study were observed to be significantly higher in patients with proctitis than in patients with other types of colitis in the A-FMT group. However, this tendency was not observed in the mono-AFM group. We had previously reported that the A-FMT regimen synergistically contributed to the recovery of Bacteroidetes composition associated with clinical response and endoscopic severity (endoscopic sum score) [23,24]. The findings of the present study strengthen our previous results. These findings suggest that microbiota transplantation will not be successful if the patient’s intestinal tract does not have sufficient mucins for bacterial colonization because Bacteroidetes species may be unable to adhere to the niche of the extensively damaged mucosa, which might have lost essential components related to intestinal immunity [33]. Moreover, it has been reported that intestinal IgA in the mucin layer is important for the control of commensal gut microbiota and that the selective action of intestinal bacteria by IgA may be related to the mechanism of bacterial colonization [34].

Further, regarding the long-term clinical evaluation of A-FMT, to the best of our knowledge, this is a valuable report to highlight the importance of donor selection for FMT in UC. In the present study, the noteworthy observations related to the microbiota are as follows: (1) cumulative non-relapse rate in the long-term was significantly higher in the A-FMT group than in the mono-AFM group; and (2) matching between donors (e.g., siblings, close in age) and patients is an essential factor for long-term maintenance. However, this trend was not observed during short-term evaluation. Previously, we have reported that eradication of indigenous bacterial microbiota by AFM pre-treatment may promote the effective colonization of Bacteroidetes species supplied by FMT [23,24]. AFM therapy may enhance the long-term maintenance of FMT by eradicating pathogenic bacteria and resetting the dysbiotic intestinal microbiota while minimizing the recipient microbiota, thereby facilitating the establishment of exogenous microbiota. FMT may also reverse antimicrobial-associated dysbiosis caused by AFM therapy, thereby enhancing the efficacy of the treatment. Recent studies have demonstrated that the choice of the donor does not influence the clinical efficacy of FMT in comprising CDI patients [35,36]. On the contrary, the microbial diversity of donors is a major factor influencing the efficacy of FMT in UC patients [12,37]. Two reasons may explain this difference from our findings. One plausible reason for this difference is the criteria for donor selection. Most studies on FMT have employed non-relatives as donors for convenience of delivery to many patients, whereas we recruited the patients’ relatives as donors. This approach for donor selection can reveal the importance of matching between donors and recipients. Another possible reason for the difference is the administration of multiple antibiotics during the pre-treatment period. The traditional method for repeated FMT (intensive FMT) is a way to induce therapeutic effects by raising diversity and operational taxonomic units. In A-FMT, AFM antibiotic pre-treatment resets its microbiota, and matching of microbiota is important for whole microbiota transplantation [9,38]. Therefore, using a single donor is more reasonable than employing multiple donors in A-FMT.

It is interesting to note that the short-term efficacy of A-FMT is not associated with the type of donors and that donor selection can affect long-term maintenance and prevent relapse in A-FMT. “Colonization” of microbiota may be different from its “stabilization” such that the long-term maintenance rate is affected in UC. In other words, colonization is not associated with the type of donors, whereas stabilization of microbiota depends on the donors. Genetic and immunologic factors may influence the stabilization of microbiota. Related healthy individuals are known to harbor similar gut microbiota [39]. Some studies have reported that the acquisition of gut microbiota predominantly occurs over the first few years of life because the epidemiological associations of IBD are linked to the early childhood stage (including delivery, breastfeeding, weaning diet, maternal inoculum, and home environment) and subsequent antibiotic exposures [40,41,42]. The gut microbiota of siblings may reflect microbiota prior to UC development in patients, although we employed relatives as donors with a risk for developing UC because genetic susceptibility is also an important factor for IBD. Furthermore, the results of our study suggest that donors with less difference in age had acceptable microbiota to the patients. Of note, in cases #4, #25, and #26, the elderly patients received feces from elderly donors. Nevertheless, some reports have suggested that the loss of gut microbiota diversity affected the aging process [43,44]. In other words, the required gut microbiota for the recipient may be different depending on the age group of the donor. Non-relative donors may lead to the same efficacy of FMT if the microbiota type is close to the recipient’s microbiota. These findings may provide an initial step in more efficiently stratified FMT depending upon the donor and the recipient.

In long-term evaluation, donor-microbiota stabilization was observed in case #11, wherein, the gut microbiota of recipient was similar to the donor’s microbiota (Figure 6a). The dendrogram showed a high similarity between gut microbiota of the recipient after A-FMT and the donor, thereby indicating that it was stabilized (Figure 6b). However, this finding was not observed clearly in other cases. This fact might suggest the amelioration of clinical indices in other cases without donor-microbiota stabilization, or this is suggestive of the complicated mechanism of the microbial establishment by creating a balance between the recovery of self-microbiota and colonized-microbiota, which struggle with each other [45,46]. Establishment of microbiota can also be affected by specific dietary features and/or antibiotics, as has been reported previously [47,48,49]. It would be difficult to analyze long-term changes in microbiota by minimalizing influences from these factors.

Further, we discuss the beneficial Bacteroidetes species in UC. We have previously reported a decrease in beneficial Bacteroidetes species in UC patients [24]. We found that some specific Bacteroidetes species correlated with the long-term efficacy of FMT, though we analyzed microbiota of a limited number of patients for the long term in A-FMT. These species are known to be predominant in both healthy donors and UC patients [24]. The key bacteria may differ among individuals. Costello et al. showed that anaerobically prepared donor FMT resulted in a higher likelihood of remission [15]. Bacteroidetes are also anaerobic bacteria. The results reported by us are in agreement with other studies [15,50]. Though important bacterial species and metabolites have been identified in various reports [18,51], further evaluation of the intestinal environment is required to understand the interactions among them.

Our study has some limitations. First, this study was non-randomized and conducted at a single center. Second, the clinical response was evaluated only by clinical score. A clinical study, including a randomized controlled trial, is required to evaluate the efficacy of A-FMT regimen.

## 5. Conclusions

The simple regimen called A-FMT exhibited long-term efficacy as compared to mono-AFM regimen, and it may serve as a useful strategy for managing and maintaining UC. To the best of our knowledge, this is the first report showing the importance of matching between donors and recipients for long-term maintenance by A-FMT in UC. The donor-recipient matching may help to plan the FMT regimen in real clinical practice effectively.


**Conference presentation:**
15th Congress of ECCO2020 on 12–15 February 2020, in Vienna, Austria. This study won the prize for “Top Ten Best DOPs” during the ECCO2020 conference.2020 Crohn’s & Colitis Congress on 23–25 January 2020, in Austin, Texas, USA.61st Annual Meeting of the Japanese Society of Gastroenterology on 21–24 November 2019, in Kobe, Japan.


## Figures and Tables

**Figure 1 jcm-09-01650-f001:**
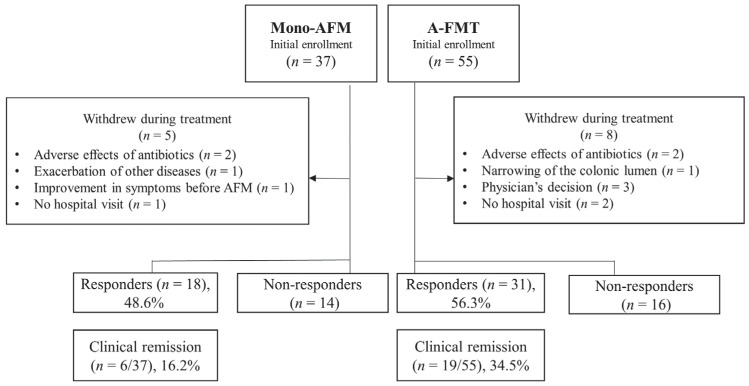
Flow diagram of this study for short-term evaluation. Ninety-two patients were enrolled in our study: 37 patients in the mono-AFM (triple antibiotic therapy (amoxicillin, fosfomycin, and metronidazole)) group and 55 patients in the A-FMT (fresh fecal microbiota transplantation (FMT) following AFM) group were assessed for clinical response and remission using intention-to-treat analysis. In the A-FMT group, 31 patients (56.3%) showed a clinical response and 19 (34.5%) achieved clinical remission. These rates were higher than the rates in the mono-AFM group (clinical response/remission: *n* = 18/6, 48.6%/16.2%), but the difference was not statistically significant.

**Figure 2 jcm-09-01650-f002:**
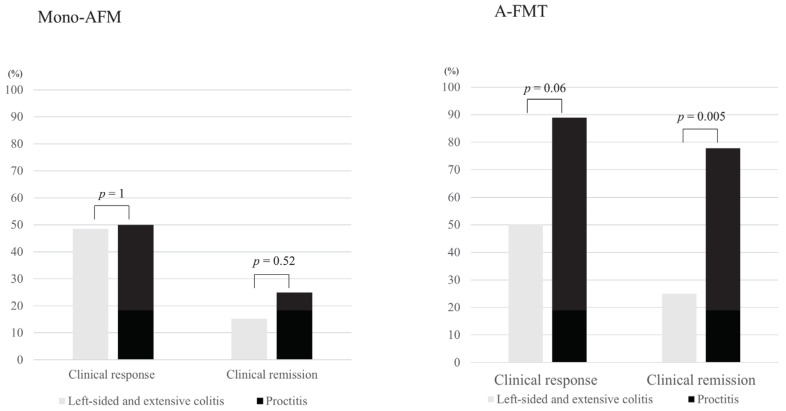
Comparison of the clinical response rate and remission rate by disease distribution between the mono-AFM and A-FMT groups during short-term evaluation. Although no significant difference was observed in the mono-AFM (triple antibiotic therapy (amoxicillin, fosfomycin, and metronidazole)) group, the response and remission rates were significantly higher in patients with proctitis than in those with left-sided and extensive colitis in the A-FMT (fresh fecal microbiota transplantation (FMT) following triple antibiotic therapy (amoxicillin, fosfomycin, and metronidazole)) group (left-sided and extensive colitis, *n* = 46; proctitis, *n* = 9; clinical response, *p* = 0.06; clinical remission, *p* = 0.005, chi-square test).

**Figure 3 jcm-09-01650-f003:**
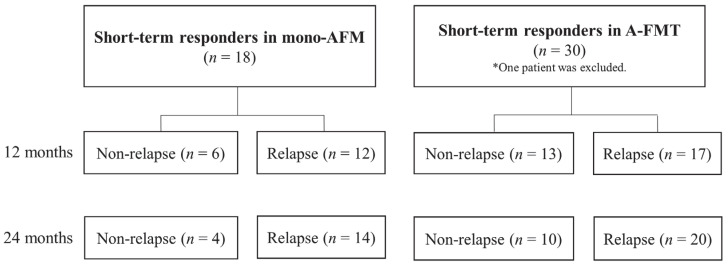
Flow diagram for long-term evaluation of patients who showed effects at 4 weeks after treatment. In the A-FMT (fresh fecal microbiota transplantation (FMT) following triple antibiotic therapy ( amoxicillin, fosfomycin, and metronidazole)) group, 13 patients did not experience relapse within 12 months, and 10 patients did not experience relapse within 24 months after receiving A-FMT. In the mono-AFM (triple antibiotic therapy (amoxicillin, fosfomycin, and metronidazole)) group, 6 patients did not experience relapse within 12 months, and 4 patients did not experience relapse within 24 months after AFM administration. * One patient in the A-FMT group was excluded because of the detection of dysplasia.

**Figure 4 jcm-09-01650-f004:**
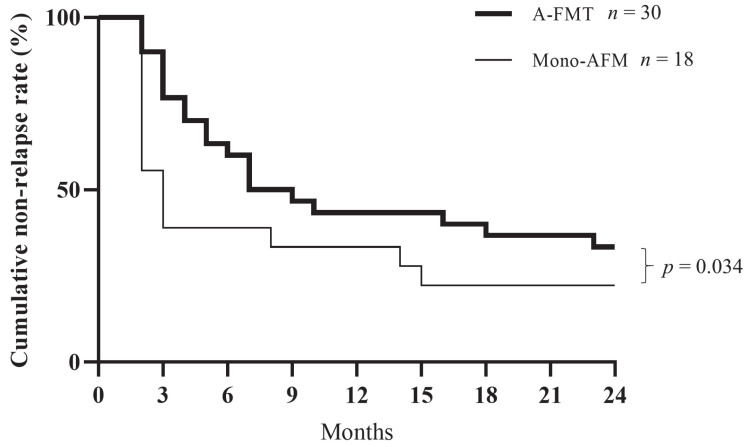
Comparison of cumulative non-relapse rate between the mono-AFM and A-FMT groups. The cumulative non-relapse rate for 24 months was significantly higher in the A-FMT (fresh fecal microbiota transplantation (FMT) following triple antibiotic therapy (amoxicillin, fosfomycin, and metronidazole)) group than in the mono-AFM (triple antibiotic therapy (amoxicillin, fosfomycin, and metronidazole)) group (*p* = 0.034, Gehan-Breslow-Wilcoxon test).

**Figure 5 jcm-09-01650-f005:**
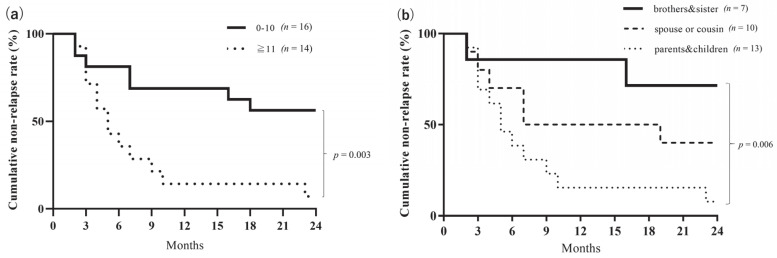
(**a**) Comparison of the cumulative non-relapse rate by the age difference between donors and patients in the A-FMT group. In the A-FMT (fresh fecal microbiota transplantation (FMT) following triple antibiotic therapy (amoxicillin, fosfomycin, and metronidazole)) group, the ≥11-year difference group (*n* = 14) showed significantly lower cumulative non-relapse rate than the 0–10-year difference group (*n* = 16) for 24 months (*p* = 0.003, log-rank test). (**b**) Comparison of cumulative non-relapse rate by the relationship between donors and patients in the A-FMT group. In the A-FMT (fresh fecal microbiota transplantation (FMT) following triple antibiotic therapy (amoxicillin, fosfomycin, and metronidazole)) group, sibling FMT (*n* = 7) had a significantly higher cumulative remission rate than parent-child FMT (*n* = 13) for 24 months (*p* = 0.007, log-rank test).

**Figure 6 jcm-09-01650-f006:**
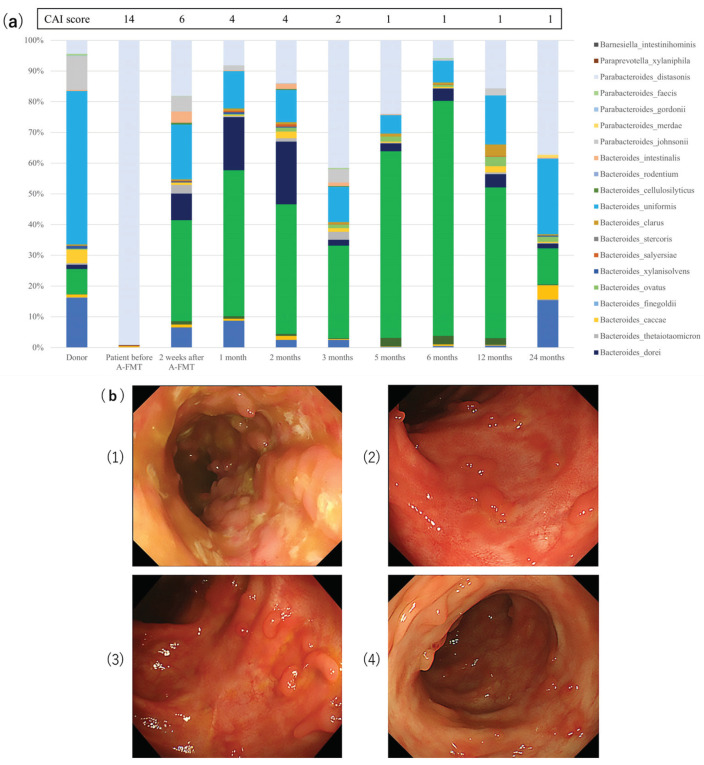
(**a**) Change in microbiota composition of a patient before and after A-FMT with respect to Bacteroidetes species as compared to the donor. In the A-FMT (fresh fecal microbiota transplantation (FMT) following triple antibiotic therapy (amoxicillin, fosfomycin, and metronidazole)) group, the change in the composition of Bacteroidetes species in 10 samples (donor, patient before FMT, at 2 weeks, 1, 2, 3, 5, 6, 12, and 24 months after FMT) in case #11. (**b**) Example of long-term endoscopic outcome after A-FMT. Case #11: A 30-year-old man with a history of extensive ulcerative colitis for three years. He did not take any medication, such as 5-aminosalicylate treatment, PSL or anti-TNFα. He received A-FMT (fresh fecal microbiota transplantation (FMT) following triple antibiotic therapy (amoxicillin, fosfomycin, and metronidazole)) from his younger brother.

**Table 1 jcm-09-01650-t001:** Comparison of the baseline characteristics of patients who received mono-AFM and A-FMT.

	All (*n* = 92)	Mono-AFM (*n* = 37)	A-FMT (*n* = 55)	*P* Value
Age (years)	41.1 ± 13.9	42.5 ± 14.7	40.1 ± 13.3	0.42
Sex (M/F)	56/36	18/19	38/17	0.049
Duration of disease (years)	8.7 ± 8.1	8.9 ± 9.0	8.6 ± 7.4	0.83
Disease location				
Proctitis	13	4	9	0.66
Left-sided colitis	27	11	16	0.95
Extensive colitis	52	22	30	0.64
Ongoing/past treatment				
5-ASA	78/86	32/35	46/51	0.94/1
Corticosteroid	30/67	14/30	16/37	0.38/0.22
Apheresis	1/38	0/19	1/19	0/0.11
Azathioprine	20/38	7/16	13/22	0.78/0.76
Tacrolimus	6/16	2/5	4/11	0.47/0.60
Anti-TNF	16/35	6/12	10/23	0.97/0.36
CAI score	10.1 ± 2.7	9.6 ± 2.3	10.3 ± 2.9	0.21
≥11	36	11	25	0.13
6–10	53	25	28	0.11
≤5	3	1	2	0.06
Endoscopic evaluation				
UCEIS	4.4 ± 2.4	3.8 ± 2.0	4.7 ± 2.5	0.12
Mayo endoscopic score	1.9 ± 0.7	1.8 ± 0.8	1.9 ± 0.7	0.86
Sum of Mayo endoscopic scores	6.3 ± 4.2	6.3 ± 4.3	6.3 ± 4.1	0.99

A-FMT: fecal microbiota transplantation (FMT) following triple antibiotic therapy (amoxicillin, fosfomycin, and metronidazole); Mono-AFM: triple antibiotic therapy (amoxicillin, fosfomycin, and metronidazole); 5-ASA: 5-aminosalicylic acid; Anti-TNF: anti-tumor necrosis factor; UCEIS: ulcerative colitis endoscopic index of severity.

**Table 2 jcm-09-01650-t002:** Baseline characteristic of responders treated with A-FMT.

Case	Age & Sex	Duration of UC (years)	Type of UC	Donor	Donor Age & Sex	Age Difference (years)	Decrease in CAI Score	Sum of Mayo Endoscopic Scores	Duration of Maintenance (months)
#1	27M	9	Extensive	Mother	59F	32	4	9	3
#2	24M	6	Extensive	Mother	48F	24	5	9	10
#3	26F	0.5	Left-sided	Father	58M	32	4	4	9
#4	62M	1	Proctitis	Spouse	58F	4	8	1	24 *
#5	32M	10	Proctitis	Spouse	29F	3	7	1	3
#6	36M	3	Proctitis	Spouse	30F	6	9	1	24 *
#7	43M	2.5	Left-sided	Spouse	43F	0	9	2	7
#8	22F	1.5	Left-sided	Father	56M	34	7	2	24 *
#9	46M	26	Extensive	Spouse	49F	3	13	10	2
#10	61F	24	Proctitis	Daughter	32F	29	3	1	5
#11	30M	3.5	Extensive	Brother	27M	3	8	12	24 *
#12	37M	6	Extensive	Spouse	35F	2	4	8	18
#13	60F	5	Extensive	Daughter	26F	34	5	5	23
#14	55F	6	Proctitis	Sister	53F	2	3	1	16
#15	65M	23	Extensive	Daughter	32F	33	3	12	4
#16	47M	27	Proctitis	Spouse	46F	1	3	4	24 *
#17	46F	5	Extensive	Sister	49F	3	5	2	24 *
#18	31M	14	Proctitis	Father	66M	33	4	1	6
#19	55F	34	Left-sided	Cousin	20F	35	13	2	4
#20	30M	12	Extensive	Mother	53F	23	6	8	3
#21	29M	5	Extensive	Mother	56F	27	7	6	7
#22	40M	3	Left-sided	Mother	65F	25	8	3	5
#23	24F	11	Left-sided	Mother	48F	24	8	3	3
#24	36M	6	Extensive	Spouse	43F	7	3	11	7
#25	64F	13	Left-sided	Sister	58F	6	11	5	24 *
#26	73F	14	Extensive	Sister	65F	8	6	7	24 *
#27	32M	12	Extensive	Sister	30F	2	12	8	2
#28	26M	5	Left-sided	Sister	25F	1	4	2	24 *
#29	43F	7	Left-sided	Spouse	46M	3	12	5	24 *
#30	33M	15	Proctitis	Father	66M	33	6	3	2

* The patients did not have a relapse for 24 months. A-FMT: fecal microbiota transplantation (FMT) following triple antibiotic therapy (amoxicillin, fosfomycin, and metronidazole); CAI: clinical activity index; UC: ulcerative colitis.

**Table 3 jcm-09-01650-t003:** Types of Bacteroidetes species that existed in donors and continued to exist in five patients who maintained without recurrence for 24 months.

	Bacteroidetes Species
Case #4	***Bacteroides uniformis*****, ***Parabacteroides distasonis*****, ***Bacteroides dorei****, *Prevotella copri*
Case #6	***Bacteroides uniformis*****, ***Parabacteroides distasonis*****, ***Bacteroides dorei****, *Alistipes putredinis*, *Alistipes onderdonkii*, *Bacteroides plebeius*, *Bacteroides coprocola*, *Bacteroides massiliensis*, *Bacteroides vulgatus*, *Bacteroides thetaiotaomicron*, *Bacteroides caccae*, *Bacteroides stercoris*, *Bacteroides rodentium*, *Parabacteroides merdae*, *Paraprevotella xylaniphila*
Case #11	***Bacteroides uniformis*****, ***Parabacteroides distasonis*****, ***Bacteroides dorei****, *Alistipes putredinis*, *Alistipes onderdonkii*, *Bacteroides massiliensis*, *Bacteroides vulgatus*, *Bacteroides thetaiotaomicron Bacteroides caccae*, *Bacteroides clarus*
Case #25	***Bacteroides uniformis*****, ***Parabacteroides distasonis*****, ***Bacteroides dorei****, *Bacteroides massiliensis*, *Bacteroides vulgatus*, *Bacteroides thetaiotaomicron*
Case #29	***Bacteroides uniformis*****, ***Parabacteroides distasonis*****, *Alistipes putredinis*, *Alistipes onderdonkii**, *Parabacteroides merdae*

** Species were observed in all five cases. * Species were observed in 4 out of 5 cases.

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
