# Peer review of "Matching between Donors and Ulcerative Colitis Patients Is Important for Long-Term Maintenance after Fecal Microbiota Transplantation"

_jcm, 2020, doi:10.3390/jcm9061650_

Round 1
Reviewer 1 Report
This paper, entitled “ Matching Between Donors and Ulcerative Colitis Patients Is Important for Long-Term Maintenance After Fecal Microbiota Transplantation” describes the analysis of the long-term efficacy of A-FMT compared to AFM monotherapy, and explores the criteria for a beneficial donor for long-term maintenance (24 months) after A-FMT.
The study is very interesting, but there are some points of concerns:
- A clear definition of mono-AFM is lacking both in the introduction and in the materials sections. In the materials & methods section a paragraph “Protocol for AFM monotherapy” should be added.
- The indication to the experimental treatment is not clear. Usually, Fecal Microbiota Transplantation is an experimental treatment reserved for those patients who are refractory to conventional treatments. The patient population of this study also included patients with mild disease (Mayo endoscopic score=1 corresponds to minimal lesions); are they resistant to standard therapies? This should be better explained. The inclusion criteria should be more detailed.
- Page 2, line 86. The treatment allocation (A-FMT or mono-AFM) is based on patient preference. What kind of information received the patients to make the choice?
- Page 4, line 148. Usually, the significance is defined by p≤0.05.
- Adverse events. The authors write that “the patients treated with AFM showed…”. Does this mean that all the patients had an adverse event? Numbers and percentages should be added, distinguishing between minor and serious adverse events.
- Table 1. Proctitis was present in 9% in the mono-AFM group and 17% of the A-FMT group. It is not clear whether this difference was significant. The p values of other pretrial characteristics are lacking.
- As far as data on clinical efficacy concerns, it is not clear whether an intention-to-treat or per-protocol analysis was performed. In figure 1 it is shown that 13 patients withdrew during treatment. According to intention-to-treat analysis, these patients should be considered as treatment failure. What kind of analysis was performed? The type of analysis should be better illustrated.
- Figure 2 has the same problems as the type of analysis.
- Figure 4 probably is the most important figure regarding clinical efficacy. However, the number of patients at risk should be added. The term “Non-relapse” should be changed in “remission”.
- In conclusion, the results section that describes the data of clinical efficacy is confusing. The results of clinical efficacy should be simplified and should be re-written more clearly.
Results about “microbial analysis” are very well described.
Author Response
Reviewer 1
- A clear definition of mono-AFM is lacking both in the introduction and in the materials sections. In the materials & methods section a paragraph “Protocol for AFM monotherapy” should be added.
Response: We thank the reviewer for a constructive comment. Per the suggestion of the reviewer, we have added a paragraph for ‘Protocol for AFM monotherapy’ in the Materials and Methods section in the revised manuscript on page 3, line 106. This paragraph has been designated by the number 2.5. Further paragraphs have been re-numbered accordingly. The following paragraph has been added:
“2.5. Protocol for AFM monotherapy
Patients received a combination antibiotic regimen consisting of oral amoxicillin (1500 mg/day), fosfomycin (3000 mg/day), and metronidazole (750 mg/day) for 2 weeks. After AFM monotherapy, the patients were permitted to continue the ongoing treatment and were regulated in terms of probiotic intake during the study period. Change in drug dosage or initiation of new treatment was not allowed.”
- The indication to the experimental treatment is not clear. Usually, Fecal Microbiota Transplantation is an experimental treatment reserved for those patients who are refractory to conventional treatments. The patient population of this study also included patients with mild disease (Mayo endoscopic score=1 corresponds to minimal lesions); are they resistant to standard therapies? This should be better explained. The inclusion criteria should be more detailed.
Response: In this study, patients who were not refractory to the conventional treatment were also invited to participate. This is because FMT is a treatment with a few side effects, and we believe it has the advantage of being used in patients who are not refractory to the conventional treatment. Per the suggestion of the reviewer, we have added details in the inclusion criteria mentioned in the Materials and Methods section on page 2, line 84 in the revised manuscript. The following sentence has been added:
“without considering the resistance to existing treatment.”
- Page 2, line 86. The treatment allocation (A-FMT or mono-AFM) is based on patient preference. What kind of information received the patients to make the choice?
Response: Per the suggestion of the reviewer, we have added an explanation for treatment allocation in the revised manuscript on page 2, line 87.
“The patients who could not find an appropriate donor candidate chose mono-AFM regimen. Additionally, some patients chose mono-AFM owing to its convenience and its proven efficacy.”
- Page 4, line 148. Usually, the significance is defined by p≤0.05.
Response: We apologize for our mistake. We have changed ‘P values ≥0.05’ to ‘P values ≤0.05’ on page 4, line 159 in the revised manuscript.
- Adverse events. The authors write that “the patients treated with AFM showed…”. Does this mean that all the patients had an adverse event? Numbers and percentages should be added, distinguishing between minor and serious adverse events.
Response: Per the suggestion of the reviewer, we have added the data for the adverse events on page 4, paragraph 3.1.
“Among 92 patients treated with AFM, 12 patients (13.0%) showed the exacerbation of diarrhea, rash, and nausea due to antibiotics; however, after the end of AFM regimen or withdrawal from oral administration (drop-out from the study), all symptoms improved and no serious drug-related toxicities were observed. Adverse events due to FMT included exacerbation of nausea and prokinetic bowel peristalsis, which were observed in 20 patients (42.6%); nevertheless, both were transient and no serious infection or diarrhea was observed.”
- Table 1. Proctitis was present in 9% in the mono-AFM group and 17% of the A-FMT group. It is not clear whether this difference was significant. The p values of other pretrial characteristics are lacking.
Response: We thank the reviewer for a constructive comment. Per suggestion, we have added P value for every characteristic in Table 1.
- As far as data on clinical efficacy concerns, it is not clear whether an intention-to-treat or per-protocol analysis was performed. In figure 1 it is shown that 13 patients withdrew during treatment. According to intention-to-treat analysis, these patients should be considered as treatment failure. What kind of analysis was performed? The type of analysis should be better illustrated.
Response: We have performed per-protocol analysis. We have mentioned this in the statistical analysis section on page 4, line 156 in the revised manuscript.
“The results were evaluated by per-protocol analysis.”
- Figure 2 has the same problems as the type of analysis.
Response: We thank the reviewer for the comment. We performed per-protocol analysis. We have mentioned this in the statistical analysis section on page 4, line 156 in the revised manuscript.
“The results were evaluated by per-protocol analysis.”
- Figure 4 probably is the most important figure regarding clinical efficacy. However, the number of patients at risk should be added. The term “Non-relapse” should be changed in “remission”.
Response: In Figure 4, 5a, and 5b, the word “Non-relapse” was used because besides the patients who achieved clinical remission, the short-term responders also included the responders (decrease of CAI ≥3). This has been described in the Materials and Methods section. Therefore, we would like to retain the original word, i.e., “Non-relapse”.
- In conclusion, the results section that describes the data of clinical efficacy is confusing. The results of clinical efficacy should be simplified and should be re-written more clearly.
Response: Per the reviewer's request, we have simplified the results of clinical efficacy by deleting the following complicated sentences on page 8, line 242 in the revised manuscript.
“In the A-FMT group, 13 (43.3%) and 10 (33.3%) patients did not experience relapse within 12 and 24 months, respectively, after undergoing A-FMT. In the mono-AFM group, 6 (33.3%) and 4 patients (22.2%) did not experience relapse within 12 and 24 months, respectively, after AFM administration.”
Reviewer 2 Report
The manuscript " Matching Between donors and Ulcerative colitis (UC) patients is important for long-term maintenance after fecal microbiota transplant (FMT)" is an important manuscript to determine the long-term effects of FMT for UC as the authors previously published the short-term effects of this research. It was a pleasure to review this paper.
This study is an interesting study which examined the efficacy of AFM and A-FMT on UC using clinical data as well as laboratory findings.
Overall, this study was not a randomized study, therefore there may be biased involved in the clinical data.
There are a few minor issues in the manuscript.
Please see them as follows:
Please consider revising highlighted areas in the attached manuscript.

Author Response
Reviewer 2
Point 1. Page3, line 98; If these positive -excluded or not?
Response: We thank the reviewer for a constructive comment. We excluded the donor candidate if the screening test was positive. We have added the following sentence in the revised manuscript on page 3, line 104.
“If this screening test was positive, the donor candidate was excluded from the study.”
Point 2. Page 3, line 119
Response: Per the suggestion of the reviewer, we have restructured the sentence in the revised manuscript on page 3, line 129, as follows:
“In the A-FMT group, colonoscopy was conducted at the time of FMT following the AFM administration.”
Point 3. Page 3, line 126; Not clear
Response: As pointed out by the reviewer, We have edited the sentence to make it clear in the revised manuscript on page 3, line 136. The sentence has been edited to: “For the long-term evaluation of short-term responders,”
Point 4. Page 10, line 287; This figure is not self explanatory. Please modify or remove it.
Response: In accordance with the reviewer's suggestion, we have removed Figure 6b and deleted the corresponding sentences from the figure legend and the text in the Results section. Following this revision, figure 6 (c) has been changed to figure 6 (b).
Point 5. Page 10, line 298; Remove italic
Response: We apologize for our mistake. We have removed the italic per the reviewer’s suggestion in the revised manuscript on page 10, line 311.
Point 6. Page 10, line 316; Please revise this sentence.
Response: Per the suggestion of the reviewer, we have revised the sentence “Important observations with respect to microbiota noted in the study” to “In the present study, the noteworthy observations related to the microbiota” in the revised manuscript on page 11, line 329.
Round 2
Reviewer 1 Report
- The authors specified that the experimental treatment was used as conventional therapy. None of the patients were resistant to conventional treatment. Current guidelines indicate that Fecal Microbiota Transplantation is an experimental treatment reserved for those patients who are refractory to conventional treatments.
The authors should add in the discussion comment on this fact –i.e. the use outside current indications. Why fecal transplantation before optimization of conventional treatment? In what kind of patients?
- The authors used the per-protocol analysis was used. This analysis includes only those patients who completed the treatment. Patients that do not complete the treatment don’t influence the results, even if they are a treatment failure. This leads to bias. Real data on treatment efficacy are generally derived from the intention-to-treat analysis that includes all patients as originally allocated after the start of therapy. A patient who does not complete the treatment should be considered a treatment failure and this doesn’t occur in the per-protocol analysis. Out of 55 A-FMT patients, 31 were short term responders (65%). At 24 months those who didn’t relapse were 10 patients (<20%).
Summarizing the conclusion of the paper about the long-term efficacy of MT should be more cautious.
Author Response
Responses to the comments of the Reviewers
Reviewer 1
- The authors specified that the experimental treatment was used as conventional therapy. None of the patients were resistant to conventional treatment. Current guidelines indicate that Fecal Microbiota Transplantation is an experimental treatment reserved for those patients who are refractory to conventional treatments. The authors should add in the discussion comment on this fact –i.e. the use outside current indications. Why fecal transplantation before optimization of conventional treatment? In what kind of patients?
Response: We thank the reviewer for this critical comment. We would like to mention that we included patients who were refractory to conventional treatments in our study. Almost all patients had already received some sort of standard treatment as shown in Table 1 of the manuscript. However, some patients who had a side effect or an allergy for 5-aminosalicylates and/or biologics were included because their remission was difficult with conventional treatments. We have added this explanation in the ‘Materials and Methods’ section under the subsection ‘Patients’ on page 2, line 84 in the revised manuscript. The following sentence has been added:
“This study included patients with UC who did not achieve clinical remission because of refraction to adequate treatment, including 5-aminosalicylates (5-ASA), immunomodulators, corticosteroids, and/or biologics. In addition, patients with allergies and/or side effects to 5-ASA and biologics were also included, even if their UC was mild.”
- The authors used the per-protocol analysis was used. This analysis includes only those patients who completed the treatment. Patients that do not complete the treatment don’t influence the results, even if they are a treatment failure. This leads to bias. Real data on treatment efficacy are generally derived from the intention-to-treat analysis that includes all patients as originally allocated after the start of therapy. A patient who does not complete the treatment should be considered a treatment failure and this doesn’t occur in the per-protocol analysis. Out of 55 A-FMT patients, 31 were short term responders (65%). At 24 months those who didn’t relapse were 10 patients (<20%). Summarizing the conclusion of the paper about the long-term efficacy of MT should be more cautious.
Response: Per the suggestion of the reviewer, we have now evaluated the short-term efficacy using intention-to-treat (ITT) analysis. We have revised the information for this in the revised manuscript in the following sections: ‘Abstract’ (page 1, line 24), ‘Materials and Methods’ (page 4, line 159), ‘Results’ (page 5, line 193; page 6, line 211), Figure 1 and its legend (page 5, line 199), and Figure 2 and its legend (page 6, line 221). In addition, we have revised Table 1, which includes characteristics of all participants (n=92). We have also revised the sentences in the ‘Results’ section (page 4, line 185). The following sentences have been added:
Abstract (page 1, line 24)
“The population for intention-to-treat analysis comprised 92 patients (A-FMT, n = 55; mono-AFM, n = 37). Clinical response was observed at 4 weeks post-treatment (A-FMT, 56.3%; mono-AFM, 48.6%).”
Materials and Methods (page 4, line 159)
“The results of the clinical efficacy were evaluated by intention-to-treat (ITT) analysis.”
Results
Page 4, line 185
“The clinical characteristics of these 92 patients are summarized in Table 1. The ratio of males to females was significantly higher in the A-FMT group than in the mono-AFM group.”
Page 5, line 193
“The clinical efficacy of treatment regimens for UC was evaluated by ITT analysis and estimated using the CAI score after 4 weeks of treatment. In the A-FMT group, 31 patients (56.3%) showed clinical response and 19 patients (34.5%) achieved clinical remission. These rates were higher than those observed in the mono-AFM group (clinical response/remission: n = 18/6, 48.6%/16.2%).”
Page 6, line 211
“(left-sided and extensive colitis, n = 46; proctitis, n = 9; clinical response, P = 0.06; clinical remission, P = 0.005). No significant differences in the extent of disease were observed in the mono-AFM group (left-sided and extensive colitis, n = 33; proctitis, n = 4; clinical response, P = 1; clinical remission, P = 0.52) (Figure 2).”
Legend to Figure 1 (Page 5, line 199)
“Ninety-two patients were enrolled in our study: 37 patients in the mono-AFM (triple antibiotic therapy [amoxicillin, fosfomycin, and metronidazole]) group and 55 patients in the A-FMT (fresh fecal microbiota transplantation [FMT] following AFM) group were assessed for clinical response and remission by intention-to-treat analysis. In the A-FMT group, 31 patients (56.3%) showed clinical response and 19 (34.5%) achieved clinical remission. These rates were higher than those in the mono-AFM group (clinical response/remission: n = 18/6, 48.6%/16.2%), but the difference was not statistically significant.”
Legend to Figure 2 (Page 5, line 221)
“(left-sided and extensive colitis, n = 46; proctitis, n = 9; clinical response, P = 0.06; clinical remission, P = 0.005, chi-square test).”
Furthermore, the long-term efficacy was evaluated by ITT analysis, and the findings have been added in the ‘Results’ section on page 6, line 230 in the revised manuscript. Additionally, we have deleted the non-relapse rate, calculated from short-term responders, from Figure 3 and its legend to avoid confusion with non-relapse rate calculated by ITT analysis.
“In the A-FMT group, 13 patients (23.6%; ITT analysis) did not experience relapse within 12 months and 10 (18.2%) patients did not experience relapse within 24 months after receiving A-FMT, whereas in the mono-AFM group, 6 patients (16.2%) did not experience relapse within 12 months and 4 patients (10.8%) did not experience relapse within 24 months after AFM administration.”